# *k*-Means+++: Outliers-Resistant Clustering

**Adiel Statman [1,\*], Liat Rozenberg [1,2]** 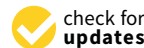 **and Dan Feldman [1]**

[1] Robotics & Big Data Lab, Computer Science Department, University of Haifa, Haifa 3498838, Israel; liatle@gmail.com (L.R.); dannyf.post@gmail.com (D.F.)

[2] School of Information and Communication Technology, Griffith University, Brisbane, QLD 4222, Australia

[\*] Correspondence: statman.adiel@gmail.com

**Abstract:** The $k$-means problem is to compute a set of $k$ centers (points) that minimizes the sum of squared distances to a given set of $n$ points in a metric space. Arguably, the most common algorithm to solve it is $k$-means++ which is easy to implement and provides a provably small approximation error in time that is linear in $n$. We generalize $k$-means++ to support outliers in two sense (simultaneously): (i) nonmetric spaces, e.g., M-estimators, where the distance $\mathrm{dist}(p, x)$ between a point $p$ and a center $x$ is replaced by $\min\{\mathrm{dist}(p, x), c\}$ for an appropriate constant $c$ that may depend on the scale of the input. (ii) $k$-means clustering with $m \geq 1$ outliers, i.e., where the $m$ farthest points from any given $k$ centers are excluded from the total sum of distances. This is by using a simple reduction to the $(k + m)$-means clustering (with no outliers).

**Keywords:** clustering; approximation; outliers

## 1. Introduction

We first introduce the notion of clustering, the solution that is suggested by $k$-means++, and the generalization of the problem to support outliers in the input. We then describe our contribution in this context.

### 1.1. Clustering

For a given similarity measure, clustering is the problem of partitioning an input set of $n$ objects into subsets, such that objects in the same group are more similar to each other than to objects in the other sets. As mentioned in [1], clustering problems arise in many different applications, including data mining and knowledge discovery [2], data compression and vector quantization [3], and pattern recognition and classification [4]. However, for most of its variants, it is an NP-Hard problem when the number $k$ of clusters is part of the input, as elaborated and proved in [5,6]. In fact, the exponential dependency in $k$ is unavoidable even for approximating the (regular) Euclidean $k$-means for clustering $n$ points in the plane [7,8].

Hence, constant or near-constant (logarithmic in $n$ or $k$) multiplicative factor approximations to the desired cost function were suggested over the years, whose running time is polynomial in both $n$ and $k$. Arguably, the most common version in both academy and industry is the $k$-means cost function, where the input is a set of $n$ points in a metric space, and the goal is to compute the set of $k$ centers (points) that minimizes the sum of squared distances over each input point to its closest center.

The $k$-means++ method that was suggested independently by [9,10] is an extremely common and simple solution with probable approximation bounds. It provides an $\alpha \in O(\log k)$ multiplicative approximation in time $O(dnk)$, e.g., for the Euclidean $d$-dimensional space. It was also shown in [10] that the approximation factor is $\alpha = O(1)$, if the input data is well separated in some formal sense.

Recently, Ref. [11] provided an $\alpha \in O(1)$ approximation for points in the $d$-dimensional Euclidean space via exactly $k$ centers in time complexity of $O(dnk^2 \log \log k)$.

The $k$-means++ algorithm is based on the intuition that the centroids should be well spread out. Hence, it samples $k$ centers iteratively, via a distribution that is called $D^2$-sampling and is proportional to the distance of each input point to the centers that were already chosen. The first center is chosen uniformly at random from the input.

The $k$-means++ algorithm supports any distance to the power of $z \geq 1$ as explained in [9]. Deterministic and other versions for the sum of (nonsquared) distances were suggested in [12–14].

A more general approximations, called *bicriteria* or $(\alpha, \beta)$ approximations, guarantee multiplicative factor $\alpha$-approximation but the number of used center (for approximating the optimal $k$ centers) is $\beta k$ for some $\beta > 1$. The factors $\alpha$ and $\beta$ might be depended on $k$ and $n$, and different methods give different dependencies and running times. For example, Ref. [15] showed that sampling $O(k \log k)$ different randomized centers yields an $O(1)$-approximation and leverage it to support streaming data. Similarly, it was proved in [16] that $D^2$-sampling $O(k)$ centers yields an $O(1)$ approximation.

The analysis of [17] explores the value of $\alpha$ as a function of $\beta$.

A coreset for the $k$-median/mean problem is a small weighted set (usually subset) of the input points that approximates the sum of distances or sum of squared distances from the original (big) set $P$ to *every* given set of $k$ centers, usually up to $(1 + \varepsilon)$ multiplicative factor. In particular, we may compute an $\alpha$-approximation on the coreset to obtain $\alpha(1 + \varepsilon)$-approximation for the original data.

For the special case of $k$-means in the Euclidean space we can replace $d$ by $O(k/\varepsilon^2)$, including for coresets constructions, as explained in [18]. Deterministic coresets for $k$-means of size independent of $d$ were suggested in [19].

*1.2. Seeding*

As explained in [10], Lloyd [20–22] suggested a simple iterative heuristic that aims to minimize the clustering cost, assuming a solution to the case $k = 1$ is known. It is a special case of the Expected Maximization (EM) heuristic for computing a local minimum. The algorithm is initialized with $k$ random points (seeds, centroids). At each iteration, each of the input points is classified to its closest centroid. A new set of $k$ centroids is constructed by taking the mean (or solving the problem for $k = 1$, in the general case) of each of the current $k$ clusters. This method is repeated until convergence or any given stopping condition.

Due to its simplicity, and the convergence to a local minimum [23], this method is very common; see [3,24–29] and references therein. The method has further improved in [1,30–32].

The drawback of this approach is that it converges to a local minimum—the one which is closest to the initial centers that had been chosen and may be arbitrarily far from the global minimum. There is also no upper bound for the convergence rate and number of iterations. Therefore, a lot of research has been done to choose good initial points, called "seeds" [33–39]. However, very few analytical guarantees were found to prove convergence.

A natural solution is to use provable approximation algorithms such as $k$-means++ above, and then apply Lloyd's algorithm as a heuristic that hopefully improves the approximation in practice. Since Lloyd's algorithm can only improve the initial solution, the provable upper bound on the approximation factor is still preserved.

*1.3. Clustering with Outliers*

In practice, data sets include some noise measurements which do not reflect a real part of the data. These are called *outliers*, and even a single outlier may completely change the optimal solution that is obtained without this outlier. One option to handle outliers is to change the distance function to a function that is more robust to outliers, such as $M$-estimators, e.g., where the distance $\text{dist}(p, x)$ between a pair of points is replaced by $\min \{\text{dist}(p, x), c\}$ for some fixed $c > 0$ that may depend on the scaling or spread of the data. Another option is to compute the set of $k$ centers that minimizes the

objective function, excluding the farthest $m$ points from the candidate $k$ centers. Here, $m \geq 1$ is a given parameter for the number of outliers. Of course, given the optimal $k$ centers for this problem, the $m$ outliers are simply the farthest $m$ input points, and given these $m$ outliers the optimal solution is the $k$-means for the rest of the points. However, the main challenge is to approximate the global optimum, i.e., compute the optimal centers and outliers simultaneously. Guessing $m$ is less challenging since it is an integer between 1 and $n$, and the cost function is monotonic in $m$, which enables the usage of binary search.

As explained in [40], detecting the outliers themselves is also an NP-hard problem [41]. Intensive research has been done on this problem, as explained in [42], since the problem of outliers is known in numerous applications [43,44]). In the context of data mining, Ref. [45] proposed a definition of distance-based outlier, which is free of any distributional assumptions and it can be generalized to multidimensional datasets. Following [45], further variations have been proposed [46,47]. Consequently, Ref. [48] introduced paradigm of local outlier factor (LOF). This paradigm has been extended in [43,49] in different directions.

As explained in [50], and following the discussion in [51,52] provided an algorithm based on Lagrange-relaxation technique. Several algorithms [51,53,54] were also developed. The work of [55] gives a factor of $O(1)$ and a running time of $O(n^m)$ for the problem of $k$-means with $m$ outliers. Another heuristic was developed by [56]. Finally, [50] provided an $O(1)$-approximation in $O(k^2(k + m)^2 n^3 \log n)$ time. In the context of $k$-means, Ref. [57] provided several algorithms of such constant factor approximation. However, the number of the points which approximate the outliers is much greater than $m$, and is dependent on the data, as well as the algorithm running time.

## 2. Our Contribution

A natural open question is: "can we generalize the $k$-means++ algorithm to handle outliers"? This paper answers this question affirmably in three senses that may be combined together:

(i) A small modification of $k$-means++ that supports $M$-estimators for handling outliers, with similar provable guarantees for both the running time and approximation factor. This family of functions includes most of the $M$-estimators, including nonconvex functions such as $M(x) = \min \{x, c\}$ for some constant $c$, where $x = \text{dist}(p, c)$ is the distance between an input point and its closest center $c$. In fact, our version in Algorithm 1 supports any pseudodistance function or $\rho$-metric that approximates the triangle inequality, as formalized in Definition 1 and Theorem 1.

(ii) A generalization of this solution to the case of $k$-mean/median problem with $m$ outliers that takes time $O(n)$ for constants $k$ and $m$. To our knowledge, this is the first nontrivial approximation algorithm that takes time linear or even near-linear in $n$. The algorithm supports all the pseudodistance functions and $\rho$-metric spaces, including the above $M$-estimators. See Corollary 4.

(iii) A Weak coreset of size $O(k+m)$ and a larger strong coreset for approximating the sum of distances to *any* $k$-centers ignoring their farthest $m$ input points. For details and exact definition see next subsection and Theorem 2.

**Novel reduction: from $k$-means to $(k+m)$-means.** While the first result is a natural generalization of the original $k$-means++, the second result uses a simple but powerful general reduction from $k$-means with $m$ outliers to $(k+m)$-means (without outliers), that we did not find in previous papers. More precisely, in Section 5, we prove that an approximation to the $(k+m)$-median with appropriate positive weight for each center (the size of its cluster), can be used to approximate the sum of distances from $P$ to *any* $k$ centers, excluding their farthest $m$ points in $P$; see Corollary 3. This type of reductions may be considered as "coreset" and we suggest two types of them.

In the first coreset, the approximation error is additive, although the final result is a multiplicative constant factor. Nevertheless, the size of the suggested coreset is only $O(k+m)$, i.e., independent of both $n$ and $d$, for constant $\rho$-metric as the Euclidean space; see Theorem 2 for details. In particular, applying exhaustive search (in time that is exponential in $k$) on this coreset implies an $O(\log k +$

$m$)-factor approximation for the $k$-median with $m$ outliers, for any $(P, \rho)$-metric in $O(n)$ time when the parameters $m$ and $k$ are constants; see Corollary 4.

As stated in the previous section, the exponential dependency in $k$ is unavoidable even for the (regular) $k$-means in the plane [7,8]. Nevertheless, constant factor approximations that take time that is polynomial in both $k$ and $m$ may be doable by applying more involved approximation algorithms for the $k$-median with $m$ outliers on our small "coreset" which contains only $O(k + m)$ points. e.g., the polynomial time algorithm of Ke Chen [50].

Theorem 2 also suggests a second "traditional coreset" that yields $(1 + \varepsilon)$-approximation for the $k$-median with $m$ outliers, i.e., that obtains $(1 + \varepsilon)$-approximation for the sum of distances from *any* set of $k$ centers and their farthest $m$ outliers in the original set $P$. The price is that we need $\alpha < \varepsilon$ approximation to the $(k + m)$-median. As was proved in [58], this is doable in the Euclidean $d$-dimensional space by running $k$-means++ for $(1/\varepsilon)^{O(d)}(k + m) \log n$ iterations instead of only $O(k + m)$ iterations (number of resulting centers). It was also proved in [58] that the exponential dependency on $d$ is unavoidable in the worst case. See Section 5 for details.

For the special case of $k$-means in the Euclidean space we can replace $d$ by $O(k/\varepsilon^2)$, including for the coresets constructions above, as explained in [18]. Deterministic version of our coresets for $k$-median with $m$ outliers may be obtained via [19].

## 3. Triangular Calculus

The algorithms in this paper support a large family of distance and nondistance functions. To exactly define this family, and their dependency on both the approximation factors and running times of the algorithms, we need to introduce the notion of $\rho$-metric that generalizes the definition $(P, \text{dist})$ of a metric space. For a point $p \in P$ and a set $X \subseteq P$, we define $f(p, X) = \min_{x \in X} f(p, x)$. The minimum or sum over an empty set is defined to be zero.

**Definition 1** ($\rho$-metric). *Let $\rho \geq 1$, $P$ be a finite set and $f : P^2 \to [0, \infty)$ be a symmetric function such that $f(p, p) = 0$ for every $p \in P$. The pair $(P, f)$ is a $\rho$-metric if for every $x, y, z \in P$ the following "approximated" triangle inequality holds:*

$$f(x, z) \leq \rho\big(f(x, y) + f(y, z)\big). \tag{1}$$

For example, metric space $(P, \text{dist})$ is a $\rho$-metric for $\rho = 1$. If we define $f(x, y) = (\text{dist}(x, y))^2$, as in the $k$-means case, it is easy to prove (1) for $\rho = 2$.

The approximated triangle inequality also holds for sets as follows.

**Lemma 1.** *(See 6.1 of [59]) Let $(P, f)$ be a $\rho$-metric. Then, for every pair of points $q, x \in P$ and a subset $X \subseteq P$ we have*

$$f(x, X) \leq \rho(f(q, x) + f(q, X)). \tag{2}$$

**Proof.** For every $p, q \in P$, and a center $x_q \in X$ that is closest to $q$, i.e., $f(q, X) = f(q, x_q)$, we have

$$f(x, X) = \min_{p \in X} f(x, p) \leq f(x, x_q) \leq \rho(f(q, x) + f(q, x_q)) = \rho(f(q, x) + f(q, X)).$$

□

Our generalization of the $k$-means++ algorithm for other distance functions needs only the above definition of $\rho$-metric. However, to improve the approximation bounds for the case of $k$-means with $m$ outliers in Section 5, we introduce the following variant of the triangle inequality.

**Definition 2** (($\rho, \phi, \varepsilon$) metric). *Let* $(P, f)$ *be a $\rho$-metric. For $\phi, \varepsilon > 0$, the pair $(P, f)$ is a $(\rho, \phi, \varepsilon)$-metric if for every $x, y, z \in P$ we have*

$$f(x, z) - f(y, z) \leq \phi f(x, y) + \varepsilon f(x, z). \tag{3}$$

For example, for a metric space $(P, \text{dist})$ the inequality holds by the triangle inequality for $\phi = 1$ and every $\varepsilon \geq 0$. For squared distance, we have $\phi = O(1/\varepsilon)$ for every $\varepsilon > 0$; see [18].

The generalization for sets is a bit more involved as follows.

**Lemma 2.** *Let $(P, f)$ be a $(\rho, \phi, \varepsilon)$-metric. For every set $Z \subseteq P$ we have*

$$|f(x, Z) - f(y, Z)| \leq (\phi + \varepsilon\rho) f(x, y) + \varepsilon\rho \min \{f(x, Z), f(y, Z)\}.$$

**Proof.** Let $z_x, z_y \in Z$ such that $f(x, Z) = f(x, z_x)$ and $f(y, Z) = f(y, z_y)$. Without loss of generality, we assume $f(y, Z) \leq f(x, Z)$. We have

$$|f(x, Z) - f(y, Z)| = f(x, Z) - f(y, Z) = f(x, z_x) - f(y, z_y) \leq f(x, z_y) - f(y, z_y)$$
$$\leq \phi f(x, y) + \varepsilon f(x, z_y), \tag{4}$$

where the last inequality is by (3).

The last term is bounded by $f(x, z_y) \leq f(y, z_y) + \phi f(x, y) + \varepsilon f(x, z_y)$ via Definition 2, but this bound is useless for the case $\varepsilon > 1$. Instead, we use (1) to obtain

$$f(x, z_y) \leq \rho(f(x, y) + f(y, z_y)) = \rho f(x, y) + \rho f(y, Z).$$

Plugging the last inequality in (4) proves the case $f(y, Z) \leq f(x, Z)$ as

$$|f(x, Z) - f(y, Z)| \leq \phi f(x, y) + \varepsilon(\rho f(x, y) + \rho f(y, Z))$$
$$= (\phi + \varepsilon\rho) f(x, y) + \varepsilon\rho f(y, Z) = (\phi + \varepsilon\rho) f(x, y) + \varepsilon\rho \min \{f(y, Z), f(x, Z)\}.$$

□

Any $\rho$-metric is also a $(\rho, \phi, \varepsilon)$-metric for some other related constants as follows.

**Lemma 3.** *Let $(P, f)$ be a $\rho$-metric. Then $(P, f)$ is a $(\rho, \phi, \varepsilon)$-metric, where $\phi = \rho$ and $\varepsilon = \rho - 1$.*

**Proof.** Let $x, y, z \in P$. We need to prove that

$$f(x, z) - f(y, z) \leq \phi f(x, y) + \varepsilon f(x, z).$$

Without loss of generality, $f(x, z) > f(y, z)$, otherwise the claim is trivial. We then have

$$f(x, z) - f(y, z) \leq \rho(f(x, y) + f(y, z)) - f(y, z) = \rho f(x, y) + (\rho - 1) f(y, z) \leq \rho f(x, y) + (\rho - 1) f(x, z),$$

where the first inequality is by the approximated triangle inequality in (1), and the second inequality is by the assumption $f(x, z) > f(y, z)$. □

How can we prove that a given function $f : P^2 \to [0, \infty)$ satisfies the condition of $\rho$-metric or $(\rho, \phi, \varepsilon)$-metric? If $f$ is some function of a metric distance, say, $f(x) = \text{dist}^r(x)$ or most M-estimators functions such as $f(x) = \min \{\text{dist}(x), 1\}$, this may be easy via the following lemma.

**Lemma 4** (Log-Log Lipschitz condition). *Let $g : [0, \infty) \to [0, \infty)$ be a monotonic nondecreasing function that satisfies the following (Log-Log Lipschitz) condition: there is $r > 0$ such that for every $x > 0$ and $\Delta > 1$ we have*

$$g(\Delta x) \leq \Delta^r g(x). \tag{5}$$

Let $(P, \text{dist})$ *be a metric space, and* $f : P^2 \to [0, \infty)$ *be a mapping from every* $p, c \in P$ *to* $f(p, c) = g(\text{dist}(p, c))$. *Then* $(P, f)$ *is a* $(\rho, \phi, \varepsilon)$-*metric where*

(i) $\rho = \max\{2^{r-1}, 1\}$,

(ii) $\phi = \left(\frac{r-1}{\varepsilon}\right)^{r-1}$ *and* $\varepsilon \in (0, r-1)$, *if* $r > 1$, *and*

(iii) $\phi = 1$ *and* $\varepsilon = 0$, *if* $r \leq 1$.

**Proof.** We denote $x = \text{dist}(p, c)$, $y = \text{dist}(q, c)$ and $z = \text{dist}(p, q)$.

(i) We need to prove that $g(z) \leq \rho(g(x) + g(y))$. If $y = 0$ then $q = c$, so

$$g(z) = g(\text{dist}(p, q)) = g(\text{dist}(p, c)) = g(x), \tag{6}$$

and Claim (i) holds for any $\rho \geq 1$. Similarly, Claim (i) holds for $x = 0$ by symmetry. So we assume $x, y > 0$. For every $b \in (0, 1)$ we have

$$
\begin{aligned}
g(z) &= g(\text{dist}(p, q)) \\
&\leq g(\text{dist}(p, c) + \text{dist}(c, q)) = g(x + y) = bg(x + y) + (1 - b)g(x + y) \\
&= bg\left(x \cdot \frac{x + y}{x}\right) + (1 - b)g\left(y \cdot \frac{x + y}{y}\right) \\
&\leq bg(x)\left(\frac{x + y}{x}\right)^r + (1 - b)g(y)\left(\frac{x + y}{y}\right)^r \\
&= (x + y)^r\left(\frac{bg(x)}{x^r} + \frac{(1 - b)g(y)}{y^r}\right),
\end{aligned}
\begin{aligned}
&\quad(7) \\
\\
&\quad(8) \\
\\
\end{aligned}
$$

where (7) is by the triangle inequality, and (8) is by substituting $x$ and $y$ respectively in (5). Substituting $b = x^r/(x^r + y^r)$ yields

$$g(z) \leq (g(x) + g(y))\frac{(x + y)^r}{x^r + y^r}. \tag{9}$$

We now compute the maximum of the factor $h(x) = \frac{(x+y)^r}{x^r + y^r}$, whose derivative is zero when

$$r(x + y)^{r-1}(x^r + y^r) - (x + y)^r \cdot rx^{r-1} = 0,$$

i.e., when $x = y$. In this case $h(x) = 2^{r-1}$. The other extremal values of $h$ are $x = 0$ and $x = \infty$ where $h(x) = 1$. Hence, $\max_{x \geq 0} h(x) = 2^{r-1}$ for $r \geq 1$ and $\max_{x \geq 0} h(x) = 1$ for $r \in (0, 1)$. Plugging these bounds in (9) yields Claim (i)

$$g(z) \leq \max\left\{2^{r-1}, 1\right\}(g(x) + g(y)).$$

(ii)–(iii) We prove that $g(x) - g(y) \leq \phi g(z) + \varepsilon g(x)$. If $y > x$ then $g(x) - g(y) \leq 0 \leq \phi g(z) + \varepsilon g(y)$. If $x \leq y$, we need to prove that $g(x) - g(y) \leq \phi g(x) + \varepsilon g(x)$. If $y = 0$ then $g(z) = g(x)$ by (6), and thus for every $\phi \geq 1$ and $\varepsilon > 0$ we have

$$g(x) - g(y) = g(x) = g(z) \leq \phi g(z) + \varepsilon g(y)$$

as desired. We thus need to prove the last inequality for $y > 0$.

We assume
$$g(x) > \phi g(z), \tag{10}$$

and $x > y$ (so $p \neq q$ and thus $z > 0$), otherwise the claim trivially holds. Let $q = \max\{r, 1\}$. Hence,

$$g(x) = g(y \cdot x/y) \leq g(y) \cdot (x/y)^r \leq g(y) \cdot (x/y)^q$$

where the first inequality is by (5) and the second is by the definition of $q$ and the assumption $x > y \geq 0$. Rearranging gives $g(y) \geq g(x) \cdot (y/x)^q$, so

$$g(x) - g(y) \leq g(x) \cdot (1 - (y/x)^q). \tag{11}$$

We now first prove that

$$g(x) \cdot (1 - (y/x)^q) \leq \varepsilon g(x) + \frac{\phi}{q^q} \cdot g(x)(1 - (y/x)^q)^q. \tag{12}$$

Indeed, if $q = 1$ then $r \leq 1$, $\varepsilon = 0$, $\phi = 1$ and (12) trivially holds with equality. Otherwise ($q > 1$), we let $p = \frac{q}{q-1}$ so that

$$
\begin{aligned}
g(x) \cdot (1 - (y/x)^q) &= (p\varepsilon g(x))^{1/p} \cdot \left( \frac{(g(x))^{1/q}(1 - (y/x)^q)}{(p\varepsilon)^{1/p}} \right) \\
&\leq \varepsilon g(x) + \frac{g(x)(1 - (y/x)^q)^q}{q(p\varepsilon)^{q/p}}
\end{aligned} \tag{13}
$$

where the inequality is by Young's inequality $ab \leq \frac{a^p}{p} + \frac{b^q}{q}$ for every $a, b \geq 0$ and $p, q > 0$ such that $1/p + 1/q = 1$. We then obtain (12) by substituting $\phi = (q-1)^{q-1}/\varepsilon^{q-1}$ so that

$$\frac{\phi}{q^q} = \frac{(q-1)^{q-1}}{q(q\varepsilon)^{q-1}} = \frac{1}{q(p\varepsilon)^{q/p}}.$$

Next, we bound the rightmost expression of (12). We have $1 - w^q \leq q(1 - w)$ for every $w \geq 0$, since the linear function $q(1 - w)$ over $w \geq 0$ is tangent to the concave function $1 - w^q$ at the point $w = 1$, which is their only intersection point. By substituting $w = y/x$ we obtain

$$1 - (y/x)^q \leq q(1 - y/x). \tag{14}$$

By the triangle inequality,

$$1 - y/x = \frac{x - y}{x} \leq \frac{z}{x}. \tag{15}$$

Combining (14) and (15) yields

$$(1 - (y/x)^q)^q \leq (q(1 - y/x))^q \leq q^q(z/x)^q. \tag{16}$$

Since $g(x) \geq \phi g(z) \geq g(z)$ by (10) and the definition of $\phi$, we have $x \geq z$, i.e., $(x/z) \geq 1$ by the monotonicity of $g$. Hence, $g(x) = g(z \cdot x/z) \leq g(z) \cdot (x/z)^r \leq g(z) \cdot (x/z)^q$ by (5), so

$$(z/x)^q \leq g(z)/g(x). \tag{17}$$

By combining the last inequalities we obtain the desired result

$$g(x) - g(y) \leq g(x) \cdot (1 - (y/x)^q) \tag{18}$$

$$\leq \varepsilon g(x) + \frac{\phi g(x)(1 - (y/x)^q)^q}{q^q} \tag{19}$$

$$\leq \varepsilon g(x) + \phi g(x)(z/x)^q \tag{20}$$

$$\leq \varepsilon g(x) + \phi g(z), \tag{21}$$

where (18) holds by (11), (19) is by (12), (20) holds by (16), and (21) is by (17). $\quad \square$

## 4. *k*-Means++ for *ρ*-Metric

In this section we suggest a generalization of the *k*-means++ algorithm for every *ρ*-metric and not only distance to the power of $z \geq 1$ as claimed in the original version. In particular, we consider nonconvex *M*-estimator functions. Our goal is then to compute an *α*-approximation to the *k*-median problem for $\alpha = O(\log k)$.

Let $w : P \to (0, \infty)$ be called the *weight function* over *P*. For a nonempty set $Q \subseteq P$ we define

$$f(Q, X) = f_w(Q, X) = \sum_{p \in Q} w(p)f(p, X).$$

If *Q* is empty then $f(Q, X) = 0$. For brevity, we denote $f(Q, p) = f(Q, \{p\})$, and $f^2(\cdot) = (f(\cdot))^2$. For an integer $k \geq 1$ we denote $[k] = \{1, \cdots, k\}$.

**Definition 3** (*k*-median for *ρ*-metric spaces). *Let* $(P, f)$ *be a ρ-metric, and* $k \in [n]$ *be an integer. A set* $X^* \subseteq P$ *is a k*-median *of P if*

$$f(P, X^*) = \min_{X \subseteq P, |X| = k} f(P, X),$$

*and this optimum is denoted by* $f^*(P, k) := f(P, X^*)$. *For* $\alpha \geq 0$, *a set* $Y \subseteq P$ *is an* *α*-approximation *for the k-median of P if*

$$f(P, X^*) \leq \alpha f^*(P, k).$$

*Note that Y might have less or more than k centers.*

The following lemma implies that sampling an input point, based on the distribution of *w*, gives a 2-approximation to the 1-mean.

**Lemma 5.** *For every nonempty set* $Q \subseteq P$,

$$\sum_{x \in Q} w(x)f_w(Q, x) \leq 2\rho f_w^*(Q, 1) \sum_{x \in Q} w(x). \tag{22}$$

**Proof.** Let $p^*$ be the weighted median of *Q*, i.e., $f_w(Q, p^*) = f_w^*(Q, 1)$. By (1), for every $q, x \in Q$,

$$f(q, x) \leq \rho(f(q, p^*) + f(p^*, x)).$$

Summing over every weighted $q \in Q$ yields

$$f_w(Q, x) \leq \rho\big(f_w(Q, p^*) + f(p^*, x) \sum_{q \in Q} w(q)\big) = \rho\big(f_w^*(Q, 1) + f(x, p^*) \sum_{q \in Q} w(q)\big).$$

Summing again over the weighted points of *Q* yields

$$\sum_{x \in Q} w(x)f_w(Q, x) \leq \rho \sum_{x \in Q} w(x)\big(f_w^*(Q, 1) + f(x, p^*) \sum_{q \in Q} w(q)\big) = 2\rho f_w^*(Q, 1) \sum_{x \in Q} w(x).$$

□

We also need the following approximated triangle inequality for weighted sets.

**Corollary 1** (Approximated triangle inequality). *Let* $x \in P$, *and* $Q, X \subseteq P$ *be nonempty sets. Then*

$$f(x, X) \sum_{q \in Q} w(q) \leq \rho(f_w(Q, x) + f_w(Q, X)).$$

**Proof.** Summing (2) over every weighted $q \in Q$ yields

$$f(x, X) \sum_{q \in Q} w(q) \le \rho \left( \sum_{q \in Q} w(q) f(q, x) + \sum_{q \in Q} w(q) f(q, X) \right)$$

$$= \rho(f_w(Q, x) + f_w(Q, X)).$$

$\square$

The following lemma states that if we use sampling from $P$ and hit a point in an uncovered cluster $Q$, then it is a 2-approximation for $f^*(P, 1)$.

**Lemma 6.** *Let* $Q, X \subseteq P$ *such that* $f_w(Q, X) > 0$. *Then*

$$\frac{1}{f_w(Q, X)} \sum_{x \in Q} w(x) f(x, X) \cdot f_w(Q, X \cup \{x\}) \sum_{q \in Q} w(q) \le 2\rho \sum_{x \in Q} w(x) f_w(Q, x).$$

**Proof.** By Corollary 1, for every $x \in Q$,

$$f(x, X) \sum_{q \in Q} w(q) \le \rho(f_w(Q, x) + f_w(Q, X)).$$

Multiplying this by $\frac{f_w(Q, X \cup \{x\})}{f_w(Q, X)}$ yields

$$f(x, X) \cdot \frac{f_w(Q, X \cup \{x\})}{f_w(Q, X)} \sum_{q \in Q} w(q) \le \rho \left( f_w(Q, x) \cdot \frac{f_w(Q, X \cup \{x\})}{f_w(Q, X)} + f_w(Q, X \cup \{x\}) \right)$$

$$\le \rho(f_w(Q, x) + f_w(Q, X \cup \{x\})) \le 2\rho f_w(Q, x).$$

After summing over every weighted point $x \in Q$ we obtain

$$\sum_{x \in Q} w(x) f(x, X) \cdot \frac{f_w(Q, X \cup \{x\})}{f_w(Q, X)} \sum_{q \in Q} w(q) \le 2\rho \sum_{x \in Q} w(x) f_w(Q, x).$$

$\square$

Finally, this lemma is the generalization of the original lemma of the *k*-means++ versions.

**Lemma 7.** *Let* $t, u \ge 0$ *be a pair of integers. If* $u \in [k]$ *and* $t \in \{0, \cdots, u\}$ *then the following holds.*

Let $\{P_1, \cdots, P_k\}$ be a partition of $P$ such that $\sum_{i=1}^{k} f^*(P_i, 1) = f^*(P, k)$. Let $U = \bigcup_{i=1}^{u} P_i$ denote the union of the first $u$ sets. Let $X \subseteq P$ be a set that covers $P \setminus U$, i.e., $X \cap P_i \ne \emptyset$ for every integer $i \in [k] \setminus [u]$, and $X \cap U = \emptyset$. Let $Y$ be the output of a call to KMEANS++$(P, w, f, X, t)$; see Algorithm 1. Then

$$E[f(P, Y)] \le \left( f(P \setminus U, X) + 4\rho^2 f^*(U, u) \right) H_t + \frac{u - t}{u} \cdot f(U, X), \tag{23}$$

where $H_t = \begin{cases} \sum_{i=1}^{t} \frac{1}{i} & \text{if } t \ge 1 \\ 1 & \text{if } t = 0 \end{cases}$, and the randomness is over the $t$ sampled points in $Y \setminus X$.

---

**Algorithm 1:** KMEANS++$(P, w, f, X, t)$; see Theorem 1.

---

**Input** : A $\rho$-metric $(P, f)$, a function $w : P \to [0, \infty)$, a subset $X \subseteq P$, and an integer
$\quad\quad\quad t \in [0, |P| - |X|]$.

**Output:** $Y \subseteq P$.

1 $Y := X$

2 **for** $i := 1 \text{ to } t$ // If $t = 0$ then skip this "for" loop

3 **do**

4 $\quad$ For every $p \in P$, $\mathrm{Pr}_i(p) = \dfrac{w(p)f(p,Y)}{\sum\limits_{q \in P} w(q)f(q,Y)}$ // $f(p, \varnothing) := 1$.

5 $\quad$ Pick a random point $y_i$ from $P$, where $y_i = p$ with probability $\mathrm{Pr}_i(p)$ for every $p \in P$.

6 $\quad$ $Y := X \cup \{y_1, \cdots, y_i\}$

7 **return** $Y$

---

**Proof.** The proof is by the following induction on $t \geq 0$: (i) the base case $t = 0$ (and any $u \geq 0$), and (ii) the inductive step $t \geq 1$ (and any $u \geq 0$).

**(i) Base Case :** $t = 0$**.** We then have

$$E[f(P,Y)] = E[f(P,X)] = f(P,X) = f(P \setminus U, X) + f(U, X) = f(P \setminus U, X) + \frac{u - t}{u} \cdot f(U, X)$$

$$\leq \left(f(P \setminus U, X) + 4\rho^2 f^*(U, u)\right) H_t + \frac{u - t}{u} \cdot f(U, X),$$

where the first two equalities hold since $Y = X$ is not random, and the inequality holds since $t = 0$, $H_0 = 1$ and $f^*(U, u) \geq 0$. Hence, the lemma holds for $t = 0$ and any $u \geq 0$.

**(ii) Inductive step:** $t \geq 1$**.** Let $y \in P$ denote the first sampled point that was inserted to $Y \setminus X$ during the execution of Line 5 in Algorithm 1. Let $X' = X \cup \{y\}$. Let $j \in [k]$ such that $y \in P_j$, and $U' = U \setminus P_j$ denote the remaining "uncovered" $u' = |\{P_1, \cdots, P_u\} \setminus P_j|$ clusters, i.e, $u' \in \{u, u - 1\}$. The distribution of $Y$ conditioned on the known sample $y$ is the same as the output of a call to KMEANS++$(P, w, f, X', t')$ where $t' = t - 1$. Hence, we need to bound

$$E[f(P,Y)] = \mathrm{Pr}(y \in P \setminus U)E[f(P,Y) \mid y \in P \setminus U] + \mathrm{Pr}(y \in U)E[f(P,Y) \mid y \in U]. \qquad (24)$$

We will bound each of the last two terms by expressions that are independent of $X'$ or $U'$.

Bound on $E[f(P,Y) \mid y \in P \setminus U]$. Here $u' = u \in [k]$, $U' = U$ (independent of $j$), and by the inductive assumption that the lemma holds after replacing $t$ with $t' = t - 1$, we obtain

$$E[f(P,Y) \mid y \in P \setminus U] \leq \left(f(P \setminus U', X') + 4\rho^2 f^*(U', u')\right) H_{t'} + \frac{u' - t'}{u'} \cdot f(U', X')$$

$$= \left(f(P \setminus U, X') + 4\rho^2 f^*(U, u)\right) H_{t-1} + \frac{u - t + 1}{u} \cdot f(U, X') \qquad (25)$$

$$\leq \left(f(P \setminus U, X) + 4\rho^2 f^*(U, u)\right) H_{t-1} + \frac{u - t + 1}{u} \cdot f(U, X).$$

Bound on $E[f(P,Y) \mid y \in U]$. In this case, $U' = U \setminus P_j$ and $u' = u - 1 \in \{0, \cdots, k - 1\}$. Hence,

$$E[f(P,Y) \mid y \in U] = \sum_{m=1}^{u} \mathrm{Pr}(j = m)E[f(P,Y) \mid j = m]$$

$$= \sum_{m=1}^{u} \mathrm{Pr}(j = m) \sum_{x \in P_m} \mathrm{Pr}(y = x)E[f(P,Y) \mid x = y]. \qquad (26)$$

Put $m \in [u]$ and $x \in P_m$. We remove the above dependency of $E[f(P, Y) \mid y \in U]$ upon $x$ and then $m$.

We have

$$
\begin{aligned}
E[f(P, Y) \mid y = x] &\leq \left(f(P \setminus U', X') + 4\rho^2 f^*(U', u')\right) H_{t-1} + \frac{u'-t'}{\max\{u', 1\}} \cdot f(U', X') \\
&= \left(f(P \setminus U, X') + f(P_m, X') + 4\rho^2 f^*(U, u) - 4\rho^2 f^*(P_m, 1)\right) H_{t-1} + \frac{u-t}{\max\{u', 1\}} \cdot f(U', X') \\
&\leq \left(f(P \setminus U, X) + 4\rho^2 f^*(U, u)\right) H_{t-1} + \frac{u-t}{\max\{u', 1\}} \cdot f(U \setminus P_m, X) \\
&\quad + \left(f(P_m, X') - 4\rho^2 f^*(P_m, 1)\right) H_{t-1},
\end{aligned}
\tag{27}
$$

where the first inequality holds by the inductive assumption if $u' \geq 1$, or since $U' = U \setminus P_j = \varnothing$ if $u' = 0$. The second inequality holds since $X \subseteq X' = X \cup \{x\}$, and since $f^*(U \setminus P_m, u - 1) = f^*(U, u) - f^*(P_m, 1)$.

Only the term $f(P_m, X') = f(P_m, X \cup \{x\})$ depends on $x$ and not only on $m$. Summing it over every possible $x \in P_m$ yields

$$
\begin{aligned}
\sum_{x \in P_m} \Pr(y = x) f_w(P_m, X') &= \frac{1}{f_w(P_m, X)} \sum_{x \in P_m} w(x) f(x, X) f_w(P_m, X \cup \{x\}) \\
&\leq \frac{2\rho}{\sum_{q \in P_m} w(q)} \sum_{x \in P_m} w(x) f_w(P_m, x) \\
&\leq \frac{2\rho}{\sum_{q \in P_m} w(q)} \cdot 2\rho f_w^*(P_m, 1) \sum_{x \in P_m} w(x) \leq 4\rho^2 f^*(P_m, 1),
\end{aligned}
$$

where the inequalities follow by substituting $Q = P_m$ in Lemma 6 and Lemma 5, respectively. Hence, the expected value of (27) over $x$ is nonpositive as

$$
\sum_{x \in P_m} \Pr(y = x)\left(f(P_m, X') - 4\rho^2 f^*(P_m, 1)\right) = -4\rho^2 f^*(P_m, 1) + \sum_{x \in P_m} \Pr(y = x) f_w(P_m, X') \leq 0.
$$

Combining this with (26) and then (27) yields a bound on $E[f(P, Y) \mid y = x]$ that is independent upon $x$,

$$
\begin{aligned}
E[f(P, Y) \mid y \in U] &= \sum_{m=1}^{u} \Pr(j = m) \sum_{x \in P_m} \Pr(y = x) E[f(P, Y) \mid y = x] \\
&\leq \left(f(P \setminus U, X) + 4\rho^2 f^*(U, u)\right) H_{t-1} + \frac{u-t}{\max\{u', 1\}} \sum_{m=1}^{u} \Pr(j = m) f(U \setminus P_m, X),
\end{aligned}
\tag{28}
$$

It is left to remove the dependency on $m$, which occurs in the last term $f(U \setminus P_m, X)$ of (28). We have

$$
\begin{aligned}
\sum_{m=1}^{u} \Pr(j = m) f(U \setminus P_m, X) &= \sum_{m=1}^{u} \frac{f(P_m, X)}{f(P, X)} \left(f(U, X) - f(P_m, X)\right) \\
&= \frac{1}{f(P, X)} \left(f^2(U, X) - \sum_{m=1}^{u} f^2(P_m, X)\right).
\end{aligned}
\tag{29}
$$

By Jensen's (or power-mean) inequality, for every convex function $g : \mathbb{R} \to \mathbb{R}$, and a real vector $v = (v_1, \cdots, v_u)$ we have $(1/u) \sum_{m=1}^{u} g(v_m) \geq g((1/u) \sum_{m=1}^{u} v_m)$. Specifically, for $g(z) := z^2$ and $v = (f(P_1, X), \cdots, f(P_u, X))$,

$$
\sum_{m=1}^{u} \frac{1}{u} f^2(P_m, X) \geq \left(\frac{1}{u} \sum_{m=1}^{u} f(P_m, X)\right)^2.
$$

Multiplying by $u$ yields

$$\sum_{m=1}^{u} f^2(P_m, X) \geq \frac{f^2(U, X)}{u}.$$

Plugging this in (29) gives the desired bound on the term $f(U', X)$,

$$\sum_{m=1}^{u} \Pr(j = m) f(U \setminus P_m, X) \leq \left(1 - \frac{1}{u}\right) \frac{f^2(U, X)}{f(P, X)} = \frac{u'}{u} \frac{f^2(U, X)}{f(P, X)} \leq \frac{\max\{u', 1\}}{u} f(U, X),$$

where the last inequality holds since $f(U, X) \leq f(P, X)$. Plugging the last inequality in (28) bounds $E[f(P, Y) \mid y \in U]$ by

$$E[f(P, Y) \mid y \in U] \leq \left(f(P \setminus U, X) + 4\rho^2 f^*(U, u)\right) H_{t-1} + \frac{u - t}{u} f(U, X). \tag{30}$$

Bound on $E[f(P, Y)]$. Plugging (30) and (25) in (24) yields

$$E[f(P, Y)] \leq \left(f(P \setminus U, X) + 4\rho^2 f^*(U, u)\right) H_{t-1}$$
$$+ f(U, X) \left(\Pr(y \in P \setminus U) \cdot \frac{u - t + 1}{u} + \Pr(y \in U) \cdot \frac{u - t}{u}\right). \tag{31}$$

Firstly, we have

$$\Pr(y \in P \setminus U) \cdot \frac{u - t + 1}{u} + \Pr(y \in U) \cdot \frac{u - t}{u} = \frac{u - t}{u} + \Pr(y \in P \setminus U) \cdot \frac{1}{u} \leq \frac{u - t}{u} + \Pr(y \in P \setminus U) \cdot \frac{1}{t},$$

where the last inequality holds as $u \geq t$.

Secondly, since $U \subseteq P$,

$$f(U, X)\Pr(y \in P \setminus U) = f(U, X) \frac{f(P \setminus U, X)}{f(P, X)} \leq f(P \setminus U, X).$$

Hence, we can bound (31) by

$$f(U, X) \left(\Pr(y \in P \setminus U) \frac{u - t + 1}{u} + \Pr(y \in U) \cdot \frac{u - t}{u}\right) \leq \frac{u - t}{u} \cdot f(U, X) + f(U, X)\Pr(y \in P \setminus U) \cdot \frac{1}{t}$$
$$\leq \frac{u - t}{u} \cdot f(U, X) + \left(f(P \setminus U, X) + 4\rho^2 f^*(U, u)\right) \cdot \frac{1}{t}. \tag{32}$$

This proves the inductive step and bounds (31) by

$$E[f(P, Y)] \leq \left(f(P \setminus U, X) + 4\rho^2 f^*(U, u)\right) H_t + \frac{u - t}{u} \cdot f(U, X).$$

□

**Corollary 2.** *Let $\delta \in (0, 1]$ and let $q_0$ be a point that is sampled at random from a nonempty set $Q \subseteq P$ such that $q_0 = q$ with probability $\dfrac{w(q)}{\displaystyle\sum_{q' \in Q} w(q')}$. Then with probability at least $1 - \delta$,*

$$f(Q, \{q_0\}) \leq \frac{2}{\delta} \rho f^*(Q, 1).$$

**Proof.** By Markov's inequality, for every non-negative random variable $G$ and $\delta > 0$ we have

$$\Pr\{G < \frac{1}{\delta} E[G]\} \geq 1 - \delta. \tag{33}$$

Substituting $G = f(Q, \{q_0\})$ yields

$$\Pr\left\{ f(Q, \{q_0\}) < \frac{1}{\delta} \sum_{q_0 \in Q} \frac{w(q_0)}{\sum_{q \in Q} w(q)} f(Q, \{q_0\}) \right\} \geq 1 - \delta. \tag{34}$$

By Lemma 5 we have

$$\sum_{q_0 \in Q} \frac{w(q_0)}{\sum_{q \in Q} w(q)} f(Q, \{q_0\}) \leq 2\rho \cdot f^*(Q, 1). \tag{35}$$

Plugging (35) in (34) yields,

$$\Pr\left\{ f(Q, \{q_0\}) < \frac{2}{\delta} \rho \cdot f^*(Q, 1) \right\} \geq \Pr\left\{ f(Q, \{q_0\}) < \frac{1}{\delta} \sum_{q_0 \in Q} \frac{w(q_0)}{\sum_{q \in Q} w(q)} f(Q, \{q_0\}) \right\}$$

$$\geq 1 - \delta.$$

□

The following theorem is a particular case of Lemma 7. It proves that the output of KMEANS++; see Algorithm 1, is a $O(\log k)$-approximation of its optimum.

**Theorem 1.** *Let $(P, f)$ be a $\rho$-metric, and $k \geq 2$ be an integer; See Definition 1. Let $w : P \to (0, \infty)$, $\delta \in (0, 1)$ and let $Y$ be the output of a call to KMEANS++$(P, w, f, \varnothing, k)$; See Algorithm 1. Then $|Y| = k$, and with probability at least $1 - \delta$,*

$$f(P, Y) \leq \frac{8\rho^2}{\delta^2} (1 + \ln k) f^*(P, k).$$

*Moreover, $|Y|$ can be computed in $O(ndk)$ time.*

**Proof.** Let $\delta' = \delta/2$ and let $\{P_1, \cdots, P_k\}$ be an optimal partition of $P$, i.e., $\sum_{i=1}^{k} f^*(P_i, 1) = f^*(P, k)$. Let $p_0$ be a point that is sampled at random from $P_k$ such that $p_0 = p$ with probability $\dfrac{w(p)}{\sum_{p' \in P_k} w(p')}$. Applying Lemma 7 with $u = t = k - 1$ and $X = \{p_0\}$ yields,

$$E[f(P, Y)] \leq \left( f(P_k, \{p_0\}) + 4\rho^2 f^*(P \setminus P_k, k - 1) \right) \cdot \sum_{i=1}^{k-1} \frac{1}{i} \tag{36}$$

$$\leq \left( f(P_k, \{p_0\}) + \frac{2\rho^2}{\delta'} f^*(P \setminus P_k, k - 1) \right) \cdot \sum_{i=1}^{k-1} \frac{1}{i} \tag{37}$$

$$= \left( f(P_k, \{p_0\}) + \frac{2\rho^2}{\delta'} f^*(P, k) - \frac{2\rho^2}{\delta'} f^*(P_k, 1) \right) \cdot \sum_{i=1}^{k-1} \frac{1}{i}, \tag{38}$$

where (38) holds by the definition of $f^*$ and $P_k$. By plugging $Q = P_k$ in Corollary 2, with probability at least $1 - \delta'$ over the randomness of $p_0$, we have

$$f(P_k, \{p_0\}) - \frac{2\rho}{\delta'} f^*(P_k, 1) \leq 0, \tag{39}$$

and since $\rho \geq 1$, with probability at least $1 - \delta'$ we also have

$$f(P_k, \{p_0\}) - \frac{2\rho^2}{\delta'} f^*(P_k, 1) \leq 0. \tag{40}$$

Plugging (40) in (38) yields that with probability at least $1 - \delta'$ over the randomness of $p_0$,

$$E[f(P,Y)] \leq \frac{2\rho^2}{\delta'} f^*(P,k) \cdot \sum_{i=1}^{k-1} \frac{1}{i} \leq \frac{2\rho^2}{\delta'} f^*(P,k) \cdot (1 + \ln k). \tag{41}$$

Relating to the randomness of $Y$, by Markov's inequality we have

$$\Pr\{f(P,Y) < \frac{1}{\delta'} E[f(P,Y)]\} \geq 1 - \delta'. \tag{42}$$

By (41) we have,

$$\Pr\left\{\frac{1}{\delta'} E[f(P,Y)] \leq \frac{2\rho^2}{\delta'^2} (1 + \ln k) f^*(P,k)\right\} \geq 1 - \delta'. \tag{43}$$

Using the union bound on (42) and (43) we obtain

$$\Pr\left\{f(P,Y) < \frac{2\rho^2}{\delta'^2} (1 + \ln k) f^*(P,k)\right\} \geq 1 - 2\delta',$$

and thus

$$\Pr\left\{f(P,Y) < \frac{8\rho^2}{\delta^2} (1 + \ln k) f^*(P,k)\right\} \geq 1 - \delta.$$

□

## 5. *k*-Means with *m* Outliers

In this section we consider the problem of $k$-means with $m$ outliers of a given set $P$, i.e., where the cost function for a given set $X$ of $k$ centers is $f(P_X, X)$ instead of $f(P, X)$, and $P_X$ is the subset of the closest $n - m$ points to the centers. Ties are broken arbitrarily. That is, $P \setminus P_X$ can be considered as the set of $m$ outliers that are ignored in the summation of errors.

**Definition 4** (*k*-median with *m* outliers)**.** *Let $(P, f)$ be a $\rho$-metric, $n = |P|$, and $k, m \in [n]$ be a pair of integers. For every subset $Q \subseteq P$ of points in $P$ and a set $X \subseteq P$ of centers, denote by $Q_X$ the closest $n - m$ points to $P$. A set $X^*$ of $k$ centers (points in $P$) is a $k$-median with $m$ outliers of $P$ if*

$$f(P_{X^*}, X^*) \leq \min_{X \subseteq P, |X|=k} f(P_X, X).$$

*For $\alpha \geq 0$, a set $Y$ is an $\alpha$-approximation to the $k$-median of $P$ with $m$ outliers if*

$$f(P_Y, Y) \leq \alpha \min_{X \subseteq P, |X|=k} f(P_X, X).$$

*Note that we allow $Y$ to have $|Y| > k$ centers.*

This is a harder and more general problem, since for $m = 0$ we get the $k$-median problem from the previous sections.

We prove in this section that our generalized $k$-means++ can be served as a "weaker" type of coreset which admits an additive error that yields a constant factor approximation for the $k$-means with $m$ outliers if $\rho$ is constant.

In fact, we prove that any approximated solution to the $(k + m)$-median of $P$ implies such a coreset, but Algorithm 1 is both simple and general for any $\rho$-distance. Moreover, by taking more than $k + m$ centers, e.g., running Algorithm 1 additional iterations, we may reduce the approximation error

$\alpha$ of the coreset to obtain "strong" regular coreset, i.e., that introduces a $(1 + \varepsilon)$-approximation for any given query set $X$ of $k$ centers. This is by having an $\alpha < \varepsilon$ approximation for the $(k + m)$ median; see Theorem 2. Upper and lower bounds for the number $|X|$ of centers to obtain such $\alpha < \varepsilon$ is the subject of [58]. The authors prove that a constant approximation to the $|X| = O((1/\varepsilon)^d k \log n)$-median of $P$ yields such $\alpha = O(\varepsilon)$-approximation to the $k$-median. This implies that running Algorithm 1 $O((1/\varepsilon)^d (k + m) \log n)$ iterations would yield a coreset that admits $(1 + \varepsilon)$-approximation error for any given set of $k$-centers with their farthest $m$ outliers, as traditional coresets. Unfortunately, the lower bounds of [58] show that the exponential dependency in $d$ is unavoidable. However, the counter example is extremely artificial. In real world data, we may simply run Algorithm 1 until the approximation error $f(P, |X|)$ is sufficiently small and hope this will happen after few $|X|$ iterations due to the structure of the data.

We state our result for the metric case and unweighted input. However, the proof essentially uses only the triangle inequality and its variants of Section 3. For simplicity of notation, we use the term multiset $C$ instead of a weighted set $(C, w)$, where $C \subseteq P$, and $w : P \to [0, \infty)$ denote the number of copies of each item in $C$. The size $|C|$ of a multiset denotes its number of points (including duplicated points), unless stated otherwise.

Unlike the case of $k$-means/median, there are few solutions to $k$ means with $m$ outliers. Chen [50] provided a (multiplicative) constant factor approximation for the $k$-median with $m$ outliers on any metric space (that satisfies the triangle inequality, i.e., with $\rho = 1$) that runs in time $O(n^3 \log(n) d k^2 (k + m)^2)$, i.e., polynomial in both $k$ and $m$. Applying this solution on our suggested coreset as explained in Corollary 3 might reduce this dependency on $n$ from cubic to linear, due to the fact that its construction time which is linear in $n$ and also polynomial in $k + m$. In order to obtain a more general result for any $\rho$-distance, as in the previous section, we use a simple exhaustive search that takes time exponential in $k + m$ but still $O(n)$ for every constant $k, m$ and $\rho$.

Our solution also implies streaming, parallel algorithm for $k$-median with $m$ outliers on distributed data. This is simply because many $k$-median algorithms exist for these computation models, and they can be used to construct the "coreset" in Theorem 2 after replacing $k$ with $k + m$. An existing offline nonparallel solution for the $k$-median with $m$ outliers, e.g., from the previous paragraph, may then be applied on the resulting coreset to extract the final approximation as explained in Corollary 4.

For every point $p \in P$, let $\text{proj}(p, Y) \in Y$ denote its closest point in $Y$. For a subset $Q$ of $P$, define $\text{proj}(Q, Y) = \{\text{proj}(p, Y) \mid p \in Q\}$ to be the union (multiset) of its $|Q|$ projection on $Y$. Ties are broken arbitrarily. Recall the definition of $P_X, C_X$ and $\alpha$-approximation from Definition 4. We now ready to prove the main technical result for the $k$-median with $m$ outliers.

**Theorem 2** (coreset for $k$-median with $m$ outliers). *Let $(P, f)$ be a $(\rho, \phi, \varepsilon)$ metric space, and let $n = |P|$. Let $k, m \in [n]$ such that $k + m < n$, and let $Y$ be an $\alpha$-approximation for the $(k + m)$-median of $P$ (without outliers). Let $C := \text{proj}(P, Y)$. Then for every set $X \subseteq P$ of $|X| = k$ centers, we have*

$$|f(P_X, X) - f(C_X, X)| \le ((\phi + \varepsilon\rho)\alpha + \varepsilon\rho) f(P_X, X), \tag{44}$$

*and*

$$f(P_X, X) \le (1 + \varepsilon\rho)f(C_X, X) + (\phi + \varepsilon\rho)\alpha f^*(P, k + m). \tag{45}$$

**Proof.** Let $X \subseteq P$ be a set of $|X| = k$ centers. The multiset $\text{proj}(P_X, Y)$ contains $n - m$ points that are contained in the $k$ centers of $Y$. However, we do not know how to approximate the number of copies of each center in $\text{proj}(P_X, Y)$ without using $P_X$. One option is to guess $(1 + \varepsilon)$ approximation to the weight of each of these $k + m$ points, by observing that it is $(1 + \varepsilon)^i$ for some $i \in O(\log(n)/\varepsilon)$, and then using exhaustive search. However, the running time would be exponential in $k + m$ and the weights depend on the query $X$.

Instead, we observe that while we cannot compute $\text{proj}(P_X, Y)$ via $C$, we have the upper bound $f(C_X, X) \leq f(\text{proj}(P_X, Y), X)$. It follows from the fact that

$$\text{proj}(P_X, Y) \subseteq \text{proj}(P, Y) = C,$$

and $|\text{proj}(P_X, Y)| = n - m$, so

$$f(C_X, X) = \min_{Q \subseteq C, |Q| = n - m} f(Q, X) \leq f(\text{proj}(P_X, Y), X). \tag{46}$$

We now bound the rightmost term.

For a single point $p \in P$ we have

$$|f(p, X) - f(\text{proj}(p, Y), X)| \leq (\phi + \varepsilon\rho)f(p, \text{proj}(p, Y)) + \varepsilon\rho \min\{f(p, X), f(\text{proj}(p, Y), X)\} \tag{47}$$

$$= (\phi + \varepsilon\rho)f(p, Y) + \varepsilon\rho \min\{f(p, X), f(\text{proj}(p, Y), X)\}, \tag{48}$$

where (47) holds by substituting $x = f(p, X)$ and $y = f(\text{proj}(p, Y), X)$ (or vice vera) and $Z = X$ in Lemma 2, and (48) holds by the definition of $f(p, Y)$.

Summing (48) over every $p \in Q$ for some set $Q \subseteq P$ yields

$$|f(Q, X) - f(\text{proj}(Q, Y), X)| = \left| \sum_{p \in Q} (f(p, X) - f(\text{proj}(p, Y), X)) \right| \tag{49}$$

$$\leq \sum_{p \in Q} |f(p, X) - f(\text{proj}(p, Y), X)| \tag{50}$$

$$\leq \sum_{p \in Q} ((\phi + \varepsilon\rho)f(p, Y) + \varepsilon\rho \min\{f(p, X), f(\text{proj}(p, Y), X)\}) \tag{51}$$

$$\leq (\phi + \varepsilon\rho)f(Q, Y) + \varepsilon\rho \min\{f(Q, X), f(\text{proj}(Q, Y), X)\}, \tag{52}$$

where (49) holds by the definition of $f$, (50) holds by the triangle inequality, and (51) is by (48). The term $f(Q, Y)$ is bounded by

$$f(Q, Y) \leq f(P, Y) \leq \alpha f^*(P, k + m), \tag{53}$$

where the first inequality holds since $Q \subseteq P$, and the last inequality holds since $Y$ is an $\alpha$-approximation for the $(k + m)$-median of $P$. Plugging (53) in (51) yields

$$|f(Q, X) - f(\text{proj}(Q, Y), X)| \leq (\phi + \varepsilon\rho)f(Q, Y) + \varepsilon\rho \min\{f(Q, X), f(\text{proj}(Q, Y), X)\}$$

$$\leq (\phi + \varepsilon\rho)\alpha f^*(P, k + m) + \varepsilon\rho \min\{f(Q, X), f(\text{proj}(Q, Y), X)\}. \tag{54}$$

The rest of the proof is by the following case analysis: (i) $f(C_X, X) \geq f(P_X, X)$, and (ii) $f(C_X, X) < f(P_X, X)$.

**Case (i):** $f(C_X, X) \geq f(P_X, X)$. The bound for this case is

$$f(C_X, X) - f(P_X, X) \leq f(\text{proj}(P_X, Y), X) - f(P_X, X) \tag{55}$$

$$\leq (\phi + \varepsilon\rho)\alpha f^*(P, k + m) + \varepsilon\rho f(P_X, X) \tag{56}$$

where (55) holds by (46), and (56) holds by substituting $Q = P_X$ in (54). This bounds (44) for the case that $f(C_X, X) \geq f(P_X, X)$.

**Case (ii):** $f(C_X, X) < f(P_X, X)$. In this case, denote the $n - m$ corresponding points to $C_X$ in $P$ by

$$\text{proj}^{-1}(C_X) := \{p \in P \mid \text{proj}(p, Y) \in C_X\}.$$

Similarly to (46),

$$\text{proj}^{-1}(C_X) \subseteq \text{proj}(P, Y) = C,$$

and $|\text{proj}^{-1}(C_X)| = n - m$, so

$$f(P_X, X) = \min_{Q \subseteq P, |Q| = n-m} f(Q, X) \le f(\text{proj}^{-1}(C_X), X). \tag{57}$$

Hence,

$$f(P_X, X) - f(C_X, X) \le f(\text{proj}^{-1}(C_X), X) - f(C_X, X) \tag{58}$$

$$\le (\phi + \varepsilon\rho)\alpha f^*(P, k+m) + \varepsilon\rho f(C_X, X) \tag{59}$$

$$\le (\phi + \varepsilon\rho)\alpha f^*(P, k+m) + \varepsilon\rho f(P_X, X), \tag{60}$$

where (58) holds by (57), (59) is by substituting $Q = \text{proj}^{-1}(C_X)$ in (54), and (60) follows from the assumption $f(C_X, X) < f(P_X, X)$ of Case (ii).

Combining all together, using (56) and (60) we can bound (44) by

$$|f(P_X, X) - f(C_X, X)| \le (\phi + \varepsilon\rho)\alpha f^*(P, k+m) + \varepsilon\rho f(P_X, X) \tag{61}$$

$$\le (\phi + \varepsilon\rho)\alpha f(P_X, X) + \varepsilon\rho f(P_X, X) \tag{62}$$

$$\le ((\phi + \varepsilon\rho)\alpha + \varepsilon\rho) f(P_X, X), \tag{63}$$

and (45) follows from (59), assuming $f(C_X, X) < f(P_X, X)$ (otherwise (45) trivially holds). $\square$

The motivation for (44) is to obtain a traditional coreset, in the sense of $(1 + \varepsilon)$-approximation to any given set of $k$ centers. To this end, we need to have $((\phi + \varepsilon\rho)\alpha) < \varepsilon$. For the natural case where $\rho = O(1)$ this implies $\alpha < \varepsilon$ approximation to the $(k+m)$-means $f^*(P, k+m)$ of $P$. This is doable for every input set $P$ in the Euclidean space and many others as was proved in [58] but in the price of taking $(1/\varepsilon)^d k \log n$ centers.

This is why we also added the bound 45, which suffices to obtain a "weaker" coreset that yields only constant factor approximation to the $k$-means with $m$ outliers, but using only $k + m$ centers, as explained in the following corollary.

**Corollary 3** (From $(k+m)$-median to $k$-median with $m$ outliers ). *Let $(P, f)$ be a $(\rho, \phi, \varepsilon)$-metric, $n = |P|$, $\alpha, \beta > 0$ and $k, m \in [n]$ such that $k + m < n$. Suppose that, in $T(n, k+m)$ time, we can compute an $\alpha$-approximation $Y$ for the $(k+m)$-median of $P$ (without outliers). Suppose also that we can compute in $t(|Y|)$ time, a $\beta$-approximation $X \subseteq C$ of the $k$-median with $m$ outliers for any given multiset $C \subseteq Y$ of $|C| = n$ (possibly duplicated) points. Then, $X$ is a*

$$(1 + \varepsilon\rho)\beta (1 + \phi\alpha + \varepsilon\rho\alpha + \varepsilon\rho) + (\phi + \varepsilon\rho)\alpha$$

*approximation for the $k$-median with $m$ outliers of $P$ and can be computed in $T(n, k+m) + t(|Y|)$ time.*

**Proof.** We compute $Y$ in $T(n, k+m)$ time and then project $P$ onto $Y$ to obtain a set $C := \text{proj}(P, Y)$ of $|Y|$ distinct points, as defined in the proof of Theorem 2. We then compute a $\beta$ approximation $X \subseteq C$ for the $k$-median with $m$ outliers of $P$,

$$f(C_X, X) \le \beta \min_{Z \subseteq C, |Z| = k} f(C_Z, Z), \tag{64}$$

in $t(|Y|)$ time. For a given set $X \subseteq P$ of centers and a subset $Q \subseteq P$ of points, we denote by $Q_X$ the closest $n - m$ points in $Q$ to $X$. Let $X^*$ denote the $k$-median with $m$ outliers of $P$. Hence,

$$f(P_X, X) \leq (1 + \varepsilon\rho)f(C_X, X) + (\phi + \varepsilon\rho)\alpha f^*(P, k + m) \tag{65}$$

$$\leq (1 + \varepsilon\rho)\beta f(C_{X^*}, X^*) + (\phi + \varepsilon\rho)\alpha f(P_{X^*}, X^*) \tag{66}$$

$$\leq (1 + \varepsilon\rho)\beta (1 + \phi\alpha + \varepsilon\rho\alpha + \varepsilon\rho) f(P_{X^*}, X^*) + (\phi + \varepsilon\rho)\alpha f(P_{X^*}, X^*) \tag{67}$$

$$\leq ((1 + \varepsilon\rho)\beta (1 + \phi\alpha + \varepsilon\rho\alpha + \varepsilon\rho) + (\phi + \varepsilon\rho)\alpha) f(P_{X^*}, X^*), \tag{68}$$

where (65) holds by (45), (66) holds by (64), and (67) holds by plugging $X^*$ in (44). □

To get rid of the so many parameters in the last theorems, in the next corollary we assume that they are all constant and suggest a simple solution to the $k$-median with $m$ outliers by running exhaustive search on our weaker coreset. The running time is exponential in $k$ but this may be fixed by running the more involved polynomial-time approximation of Chen [50] for $k$-means with $m$ outliers on our coreset.

**Corollary 4.** *Let $k, m \in [n]$ such that $k + m < n$, and let $\rho \geq 1$ be constants. Let $(P, f)$ be a $\rho$-metric. Then, a set $X \subseteq P$ can be computed in $O(n)$ time such that, with probability at least 0.99, $X$ is a $O(\ln(k + m))$-approximation for the $k$-median with $m$ outliers of $P$.*

**Proof.** Given a set $Q$ of $|Q| = n'$ points, it is easy to compute its $k$-median with $m$ outliers in $n^{O(k)}$ time as follows. We run exhaustive search over every subset $Y$ of size $|Y| = k$ in $Q$. For each such subset $Y$, we compute its farthest $m$ points $Z$ in $Q$. The set $Y$ that minimizes $f(Q \setminus Z, Y)$ is an optimal solution, since one of these sets of $k$ centers is a $k$-median with $m$ outliers of $P$. Since there are such $\binom{n'}{k} = (n')^{O(k)}$ subsets, and each check requires $O(nk) = O(n)$ time, the overall running time is $t(n') = (n')^{O(k)}$.

Plugging $\delta = 0.01$ in Theorem 1 implies that we can compute a set $Y \subseteq P$ of size $|Y| = k + m$ which is, with probability at least $1 - \delta = 0.99$, an $O(\ln(k + m))$-approximation to the $(k + m)$-median of $P$ in time $T(n, k + m) = O(nd(k + m)) = O(n)$. By Lemma 3, $(P, f)$ is a $(\rho, \phi, \varepsilon)$-metric for $\phi = \rho$, and $\varepsilon = \rho - 1$. Plugging $n' = |Y| = k$ and $\beta = 1$ in Corollary 3 then proves Corollary 4. □

## 6. Conclusions

We proved that the $k$-means++ algorithm can be generalized to handle outliers by generalizing it to $\rho$-metric that supports $M$-estimators and also proved that $k$-means seeding can be used as a coreset for the $k$-means with $m$ outliers. Open problems include generalizations of $k$-means++ for other shapes, such as $k$ lines or $k$ multidimensional subspaces, and using these approximations for developing coreset algorithms for these problems. Other directions include improving or generalizing the constant factor approximations for the original $k$-means++ and its variants in this paper, using recent improvement for the analysis of $k$-means++ and coresets for $k$-means clustering.

**Author Contributions:** Conceptualization, A.S. and D.F.; methodology, A.S. and D.F.; formal analysis, A.S. and D.F.; writing—original draft preparation, A.S. and D.F.; writing—review and editing, L.R. and D.F.; supervision, D.F.; All authors have read and agreed to the published version of the manuscript.

**Funding:** Part of the research of this paper was sponsored by the Phenomix consortium of the Israel Innovation Authority. We thank them for this support.

**Conflicts of Interest:** The authors declare no conflict of interest.

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
