# Peer review of "k-Means+++: Outliers-Resistant Clustering"

_algorithms, doi:10.3390/a13120311_

Round 1
Reviewer 1 Report
The authors proposed a variation of k-Means, named k-Means+++, which is able to deal outliers.
The proposal is well formulated but the results section is absent. Moreover, can the affirmation in Abstract "where the distance dist(p, x) between a pair of points is replaced by min {dist(p, x), 1}" be refuted if data are normalized [0,1]?
I suggest a comparison between versions of the k-Means (k-Means and k-Means++, e.g.), on scenarios for which k-Means+++ is proved be the best. How good is the proposed algorithm?
Moreover, provide a detailed methodoloy, and a statistical analisys for results can be necessary.
Author Response
We would like to thank the reviewer for the efforts spent on reading the paper, and for the encouraging review.
Comment #1: can the affirmation in Abstract "where the distance dist(p, x) between a pair of points is replaced by min {dist(p, x), 1}" be refuted if data are normalized [0,1]? Answer #1 Of course. This is similar to replacing 1 by a sufficiently large number. We can also upscale the data so that every pair would have a distance larger than 1. However, the role of this constant $c$ is to remove "far away" outliers where "far away" means more than $c$. Hence, $c$ is "scale dependent" and should be scaled with the data. To avoid the confusion, we changed the sentence to "\min {dist(p, x), c} for an appropriate data dependent constant c". Comment #2: I suggest a comparison between versions of the k-Means (k-Means and k-Means++, e.g.), on scenarios for which k-Means+++ is proved be the best. How good is the proposed algorithm?Moreover, provide a detailed methodology, and a statistical analysis for results can be necessary. Answer #2: We do not claim that our approach is better than kmeans++ but that it generalizes kmeans++ to support other distance functions, m estimators, and outliers.
Reviewer 2 Report
In this paper, the authors proved that the k-Means++ algorithm can be generalized to clustering based on any ρ-distance function and used this result to suggest a constant or near-constant factor approximations that is robust
to m outliers and takes time linear in n.
This paper is clearly written and well formulated.
If the authors number every displayed equation, some future author may be easy to reference to.
Author Response
We would like to thank the reviewer for the efforts spent on reading the paper, and for the encouraging review.
Displayed equations were numbered as requested.Reviewer 3 Report
The manuscript proposes a generalisation of k-means++ clustering algorithm to non-metric spaces and any pseudo-distance functions. This is a very interesting problem that deserves attention. The paper appears to be well-written and comprehensively referenced. I think the paper can be published in Algorithms.
Author Response
We would like to thank the reviewer for the efforts spent on reading the paper, and for the encouraging review.
Reviewer 4 Report
The paper is, in general, clearly written and very well-structured. However, I have several major concerns, regarding its content:
- What is the admissible value range for \rho? In Definition 1, no restrictions are imposed, although clearly \rho >0. In the formulation of Theorem 7, it is written that \rho should only be positive, but in the proof (the derivation of (20) from (19)) the authors claim that \rho \ge 1. The latter is again heavily used in the proof of Lemma 8. So, please, clarify the value range you consider and make sure that all your statements agree with this choice.
- There is no comparison between the results in this paper with the ones in [7]. In my opinion, the authors should compare Lemma 5 to [7, Lemma 3.4] as well as Theorem 7 to [7, Theorem 3.1] and explain the novelty of their results in detail.
- I don't see why f(y_p,Y*)\le f(y_p, M*), which is a crucial argument for the derivation of (29) and the proof of Lemma 8. Clearly, Y* being a k-median of Y is not enough, as one can choose some strange sets M and Y, such that all elements p\in M are paired with the same element y_p\in Y, which, in turn, is an outlier of Y and is closer to the k-median of M. The statement might be still true, but the close-to-optimal properties of Y and M should also be exploited (or, at least I don't see an alternative argument at the moment). Please, clarify this issue.
- There is no experimental section. In its current form, the manuscript is quite technical mathematically and the claimed advantages in terms of running time are not illustrated. Therefore, it is hard to understand and judge the achieved level of computational efficiency, making the whole paper less convincing from an application point of view.
I have also a list of minor remarks/suggestions:
- The authors should one more time go through the whole text and unify the notation and the terminology. There are places in the text, where it is obvious that another author starts writing. For example, in Lemma 8 Y and M are called k-medians, while their definition is absolutely identical to the one of k-means on page 4 (above line 110). Further, for brevity the authors decided to use the f(Q,p) notation, but in Corollary 6 and Theorem 7 they switch back to f_w(Q,{p}).
- Algorithm 1, step 4: In the denominator: "w(Y)" -> "w(q)"
- Proof of Lemma 5, the first sentences of (i) and (ii): There is no need to assume anything in the proof, as the restrictions on u and t are clearly stated in the lemma formulation.
- Lemma 8: In all the computations, based on (24) the authors use a scaling factor \rho^2\alpha instead of only \alpha. So, probably there is a typo in the formula (thus also in the lemma formulation). Again, please clarify the range of \rho and \alpha here.
- Line 143: "f(M_Y,k)" -> "f*(M_Y,k)".
- Some minor English corrections: lines 86-87: "However, the number of of the points which approximate the outliers is much greater than m, and is depend on the data, as well as the algorithm running time" -> "However, the number of the points which approximate the outliers is much greater than m, and is dependent on the data, as well as the algorithm running time"; line 102: "returns" -> "return"; lines 112-113: the sentence does not make much sense - something is missing; Lemma 5, first line: "hold" -> "holds"; line 141: "Suppose" -> "suppose"; line before 147: "from" -> "to"; Corollary 10: "be integers" -> "be an integer".
Author Response
We would like to thank the reviewer for the efforts spent on reading the paper, and for the encouraging review.
Comment #1: What is the admissible value range for \rho? In Definition 1, no restrictions are imposed, although clearly \rho >0. In the formulation of Theorem 7, it is written that \rho should only be positive, but in the proof (the derivation of (20) from (19)) the authors claim that \rho \ge 1. The latter is again heavily used in the proof of Lemma 8. So, please, clarify the value range you consider and make sure that all your statements agree with this choice. Answer #1: Fixed. Comment #2: There is no comparison between the results in this paper with the ones in [7]. In my opinion, the authors should compare Lemma 5 to [7, Lemma 3.4] as well as Theorem 7 to [7, Theorem 3.1] and explain the novelty of their results in detail. Answer #2: Our analysis generalizes k-means++ for more distance functions, e.g. those who satisfy the weak triangle inequality. We added a sentence about the factor improvement in "our contribution" in the revised version. Comment #3: I don't see why f(y_p,Y*)\le f(y_p, M*), which is a crucial argument for the derivation of (29) and the proof of Lemma 8. Clearly, Y* being a k-median of Y is not enough, as one can choose some strange sets M and Y, such that all elements p\in M are paired with the same element y_p\in Y, which, in turn, is an outlier of Y and is closer to the k-median of M. The statement might be still true, but the close-to-optimal properties of Y and M should also be exploited (or, at least I don't see an alternative argument at the moment). Please, clarify this issue. Answer #3: We modified the definition of $Y*$ in the revised version accordingly. Comment #4: There is no experimental section. In its current form, the manuscript is quite technical mathematically and the claimed advantages in terms of running time are not illustrated. Therefore, it is hard to understand and judge the achieved level of computational efficiency, making the whole paper less convincing from an application point of view. Answer #4: Correct. This is a theoretical paper and our main result is a provable approximation. However, the algorithm is very simple and similar to the kmeans++ which is why we believe it would also work in practice. The time complexity, e.g. for k-median for outliers, which analytically reduced from O(n^3 log n) to O(n), might be convincing also without experiments. Comments #5 In Lemma 8 Y and M are called k-medians, while their definition is absolutely identical to the one of k-means on page 4 (above line 110). Further, for brevity the authors decided to use the f(Q,p) notation, but in Corollary 6 and Theorem 7 they switch back to f_w(Q,{p}). Answer #5: We modify it to $k-median$ on page 4 in the revised version. The mentioned equation in this definition defines that f(Q,{p})=f_w(Q,{p})- (we put it in red in the revised version), and thus we let ourselves use both notations. Actually, we used f_w where the distinction between f(x,.) and f(X,.)=f_w(X,.) should be emphasized. In the revised version we use only f(X,.) from Lemma 5 and on. Comment #6:
- Algorithm 1, step 4: In the denominator: "w(Y)" -> "w(q)"
- Line 143: "f(M_Y,k)" -> "f*(M_Y,k)".
- Some minor English corrections: lines 86-87: "However, the number of of the points which approximate the outliers is much greater than m, and is depend on the data, as well as the algorithm running time" -> "However, the number of the points which approximate the outliers is much greater than m, and is dependent on the data, as well as the algorithm running time"; line 102: "returns" -> "return"; lines 112-113: the sentence does not make much sense - something is missing; Lemma 5, first line: "hold" -> "holds"; line 141: "Suppose" -> "suppose"; line before 147: "from" -> "to"; Corollary 10: "be integers" -> "be an integer".
Round 2
Reviewer 1 Report
The authors adjusted the paper accordding to observations reported.
Author Response
We would like to thank again the reviewer very much for the effort spent on reading the paper, on his helpful comments and on reading our answers.
Reviewer 4 Report
Unfortunately the results in the revised version of the paper are significantly weaker than the ones in the original version and, in my opinion, the contributions are not enough for an independent publication.
From the authors' answer to my major concerns it seems that the proofs of Lemma 5 and Theorem 7 follow the same ideas and techniques as the ones in [7] (Lemma 3.4 and Theorem 3.1, respectively). Improvement of a 2+ln(k) factor to a 1+ln(k) factor does not change the overall asymptotic and does not seem as a very strong result, that needs an independent publishing.
Furthermore, the new formulation and proof of Lemma 8 are not applicable in practice and still not completely correct. The new definition of Y* does not directly validate (29), since Y* is the argmin over the subsets of Y, while M* may not be such. Therefore, again some additional argumentation is needed. (I'm not saying that the statement itself is wrong, just the provided arguments are insufficient!). On top of that, the construction of the newly proposed k-median Y* explicitly uses the optimal k-mean set M, thus the set M_Y cannot be constructed in practice (it is based on Y*) and Theorem 9 and Theorem 11 are meaningless. So, such a choice of Y* is inadequate! As I said in my first revision, maybe the original choice of Y* is still ok, just the authors need to find stronger arguments or some mild additional restrictions in order to prove Lemma 8. Conducting some numerical experiments could give an indication whether such a choice could work for practical purposes, but the authors are reluctant to do them, as stated in the answer of my first revision.
Author Response
We thank the reviewer for the useful suggestions and very careful reading. These are not so common these days.
The attached version contains significant improvement and clarifications in order to answer the concerns.
The main comments of the reviewer were:
1) The results in the revised version are significantly weaker than the first version.
2) The gap from the original k-means++ paper is too small.
3) Additional argumentation is needed to prove Lemma 8.
In the attached version:
1) We add back the results from the first version, with more generalized guarantees in the context of coresets.
To this end, we reorganized the last section and divided the results to coreset and main application for $k$-median with $m$ outliers. We also put more attention to details and solve the writing issues (that lead to correction doubts) in the previous reviews. Our contribution was updated accordingly.
2) Indeed, the main ideas in Section 4 are not so novel compared to Section 5 and [7]. Nevertheless:
(a) Section 4 is mainly needed to state our main results in Section 5 for any rho-metric.
(b) We did not find the results of Section 3 in previous papers. Even the generalization for distances to the power of z>1 was not proved but sketched in a few text lines of [7]. In particular, we did not find in existing papers the exact connection between the approximation error and the chosen rho-metric.
c) To strengthen Claim (b) above, we added Section 2.
It contains new generic and non-trivial tools (especially, Lemma 4) that helped us to bound the k-means++ approximation error for non-distance non-convex functions. In particular to M-estimator for handling outliers and were not discussed in [7].
3) We found that Paper [58, new version] of Bhattacharya and Jaiswal proved recently that running k-means++ for a sufficient number of iterations yields alpha<epsilon approximation factor to k-means. This implies that we have the first traditional coreset of size independent of n for the fundamental problem of k-means with m outliers. Explanations were added in the new version. Best Regards, The Authors
Round 3
Reviewer 4 Report
The new version significantly improved all the previous version of the manuscript and the authors have thoroughly taken into consideration all my remarks. Even though I didn't have time to carefully check the new mathematical proofs (I was given only a couple of days for the revision) I am inclined to trust the authors that all the gaps are now filled and the newly provided arguments are correct. Therefore, I suggest the paper to be accepted for publication.
There are several minor remarks that should be taken into consideration. Some of them are listed below:
- line 92: There is an extra bracket after "[45-47]". Furthermore, it sounds strange that following [45] further variations have been proposed in [45] again! The authors should either change the latter to [46-47] or remove the first part of the sentence.
- The authors should carefully check the English in the newly inserted parts of the text. It needs slight improvement, in general. For example the English style of the contribution (i) on line 104 differs from the corresponding style of contributions (ii) and (iii) (there is a verb at the beginning of the sentence and it is unclear to which subject it refers to). Moreover, there are several cases where third-person singular verbs are missing their -s at the end, e.g., line 112 "support" -> "supports" and line 135 "suggest" -> "suggests".
- The enumeration style of definitions/lemmas differs from the previous two versions. Here, definitions and lemmas are unified in one family and lemma 2 follows after definition 1, while before we had both definition 1 and lemma 1. This issue depends on the general journal policy and the authors might need to double check it.
- The notation f(x,X) is firstly used in Lemma 2, but is formally defined much later at the beginning of Section 4. Thus, some reordering is needed.
- The non-original lemmas in the text are missing their references (for example, in the previous versions for Lemma 2 it was explicitly stated that it is from 6.1 in [54]). Maybe it will be better such a references to be included so that the reader can easily understand which results are original and which results are classical.
- There is a change of notation between Definition 1 and Definition 3, which I find confusing for the leader. If in definition 3 the authors change the variables x -> q, z -> x, y -> p, then the correspondence between the two definitions will be much clearer and the statement of Lemma 5 - much more easier to comprehend.
- Line 174 and below: The text needs polishing. There is probably an extra <= sign left above and the paragraph bellow starts with "Without We" which does not make much sense.
- In the statement of Theorem 15 and the following corollaries it is not explicitly written that k+m < n, as in previous versions. I agree that this is trivial to be deduced, but still maybe it is better to be written in order not to confuse the reader.
Author Response
We thank the reviewer very much for the comments and the very careful reading.
We have fixed all of the comments in the revised version and they are marked in red. We also went over the English and it is now corrected (also marked in red). Best Regards, The Authors.